# Common Variation in the PIN1 Locus Increases the Genetic Risk to Suffer from Sertoli Cell-Only Syndrome

**DOI:** 10.3390/jpm12060932

**Published:** 2022-06-04

**Authors:** Miriam Cerván-Martín, Lara Bossini-Castillo, Andrea Guzmán-Jimenez, Rocío Rivera-Egea, Nicolás Garrido, Saturnino Luján, Gema Romeu, Samuel Santos-Ribeiro, José A. Castilla, M. Carmen Gonzalvo, Ana Clavero, F. Javier Vicente, Vicente Maldonado, Sara González-Muñoz, Inmaculada Rodríguez-Martín, Miguel Burgos, Rafael Jiménez, Maria Graça Pinto, Isabel Pereira, Joaquim Nunes, Josvany Sánchez-Curbelo, Olga López-Rodrigo, Iris Pereira-Caetano, Patricia Isabel Marques, Filipa Carvalho, Alberto Barros, Lluís Bassas, Susana Seixas, João Gonçalves, Sara Larriba, Alexandra M. Lopes, F. David Carmona, Rogelio J. Palomino-Morales

**Affiliations:** 1Departamento de Genética e Instituto de Biotecnología, Centro de Investigación Biomédica, Universidad de Granada, Armilla, 18100 Granada, Spain; mcervan@ugr.es (M.C.-M.); andreeagj@correo.ugr.es (A.G.-J.); saragonmu@ugr.es (S.G.-M.); inmaculada.roma@gmail.com (I.R.-M.); mburgos@ugr.es (M.B.); rjimenez@ugr.es (R.J.); dcarmona@ugr.es (F.D.C.); 2Instituto de Investigación Biosanitaria ibs.GRANADA, 18012 Granada, Spain; josea.castilla.sspa@juntadeandalucia.es (J.A.C.); mariac.gonzalvo.sspa@juntadeandalucia.es (M.C.G.); anaclaveroglbrt@ugr.es (A.C.); fjvicenteprados.sspa@juntadeandalucia.es (F.J.V.); rpm@ugr.es (R.J.P.-M.); 3Andrology Laboratory and Sperm Bank, IVIRMA Valencia, 46015 Valencia, Spain; rocio.rivera@ivirma.com; 4IVI Foundation, Health Research Institute La Fe, 46026 Valencia, Spain; nicolas.garrido@ivirma.com; 5Servicio de Urología, Hospital Universitari i Politecnic La Fe e Instituto de Investigación Sanitaria La Fe (IIS La Fe), 46026 Valencia, Spain; satur.lujan@ivirma.com (S.L.); gema.mag@gmail.com (G.R.); 6IVI-RMA Lisbon, 1800-282 Lisbon, Portugal; samuel.sibeiro@ivirma.com; 7Department of Obstetrics and Gynecology, Faculty of Medicine, University of Lisbon, 1649-028 Lisbon, Portugal; 8Unidad de Reproducción, UGC Obstetricia y Ginecología, HU Virgen de las Nieves, 18014 Granada, Spain; 9CEIFER-GAMETIA Biobank, 18004 Granada, Spain; 10UGC de Urología, HU Virgen de las Nieves, 18014 Granada, Spain; 11UGC de Obstetricia y Ginecología, Complejo Hospitalario de Jaén, 23007 Jaén, Spain; vicenter.maldonado.sspa@juntadeandalucia.es; 12Instituto de Parasitología y Biomedicina López-Neyra, 18016 Granada, Spain; 13Centro de Medicina Reprodutiva, Maternidade Alfredo da Costa, Centro Hospitalar Universitário de Lisboa Central, 2890-045 Lisbon, Portugal; graca.pinto@chlc.min-saude.pt; 14Departamento de Obstetrícia, Ginecologia e Medicina da Reprodução, Hospital de Santa Maria, Centro Hospitalar Universitário de Lisboa Norte, 1649-028 Lisbon, Portugal; isabelsofiapereira@yahoo.com (I.P.); jjdnunes@gmail.com (J.N.); 15Laboratory of Seminology and Embryology, Andrology Service-Fundació Puigvert, 08025 Barcelona, Spain; jrsanchez@fundacio-puigvert.es (J.S.-C.); olopez@fundacio-puigvert.es (O.L.-R.); lbassas@fundacio-puigvert.es (L.B.); 16Departamento de Genética Humana, Instituto Nacional de Saúde Dr. Ricardo Jorge, 1600-609 Lisbon, Portugal; iris.caetano@insa.min-saude.pt (I.P.-C.); joao.goncalves@insa.min-saude.pt (J.G.); 17i3S—Instituto de Investigação e Inovação em Saúde, Universidade do Porto, 4200-135 Porto, Portugal; pmarques@ipatimup.pt (P.I.M.); filipac@med.up.pt (F.C.); abarros@med.up.pt (A.B.); sseixas@ipatimup.pt (S.S.); alopes@ipatimup.pt (A.M.L.); 18Institute of Molecular Pathology and Immunology of the University of Porto (IPATIMUP), 4200-135 Porto, Portugal; 19Serviço de Genética, Departamento de Patologia, Faculdade de Medicina da Universidade do Porto, 4200-319 Porto, Portugal; 20ToxOmics—Centro de Toxicogenómica e Saúde Humana, Nova Medical School, 1169-056 Lisbon, Portugal; 21Human Molecular Genetics Group, Bellvitge Biomedical Research Institute (IDIBELL), L’Hospitalet de Llobregat, 08908 Barcelona, Spain; slarriba@idibell.cat; 22Departamento de Bioquímica y Biología Molecular I, Universidad de Granada, 18071 Granada, Spain

**Keywords:** severe spermatogenic failure, male infertility, PIN1, single-nucleotide polymorphism, Sertoli cell-only syndrome

## Abstract

We aimed to analyze the role of the common genetic variants located in the *PIN1* locus, a relevant prolyl isomerase required to control the proliferation of spermatogonial stem cells and the integrity of the blood–testis barrier, in the genetic risk of developing male infertility due to a severe spermatogenic failure (SPGF). Genotyping was performed using TaqMan genotyping assays for three *PIN1* taggers (rs2287839, rs2233678 and rs62105751). The study cohort included 715 males diagnosed with SPGF and classified as suffering from non-obstructive azoospermia (NOA, *n* = 505) or severe oligospermia (SO, *n* = 210), and 1058 controls from the Iberian Peninsula. The allelic frequency differences between cases and controls were analyzed by the means of logistic regression models. A subtype specific genetic association with the subset of NOA patients classified as suffering from the Sertoli cell-only (SCO) syndrome was observed with the minor alleles showing strong risk effects for this subset (OR_add_rs2287839 = 1.85 (1.17–2.93), OR_add_rs2233678 = 1.62 (1.11–2.36), OR_add_rs62105751 = 1.43 (1.06–1.93)). The causal variants were predicted to affect the binding of key transcription factors and to produce an altered *PIN1* gene expression and isoform balance. In conclusion, common non-coding single-nucleotide polymorphisms located in *PIN1* increase the genetic risk to develop SCO.

## 1. Introduction

While mature proteins require a specific 3D structure to exert their functional activities, such complex conformation is not always determined by their amino acid sequence alone [1,2]. Actually, small proteins (less than 100 amino acids) fold autonomously, but larger proteins tend to misfold [1]. Since a higher number of long proteins are required in eukaryotes compared to prokaryotes, co-translational folding strategies have been developed and specific proteins are used to control eukaryotic protein folding and to keep protein homeostasis (known as proteostasis) [1,3]. Chaperones are generally located in the endoplasmic reticulum (ER) and interact in a fine-tuned network to prevent both protein misfolding and intermediate aggregate formation [1,3]. These intermediate aggregates turn on a specific unfolded protein response (UPR) in the ER and have been identified to play a role in a number of human diseases [3].

A major aspect of protein folding is the cis/trans conformation of the peptide bond between each proline residue and its preceding amino acid [4]. The majority of peptide bonds adopt a trans-conformation, but in the case of proline a higher proportion of cis conformations are required to maintain a proper protein structure and function [4]. The proline isomerization is a slow process with high entropy and it needs the catalyzation mediated by specific enzymes known as prolyl isomerases [4]. Moreover, the coexistence of cis/trans isomers in proteins has been suggested to be linked to regulatory processes that control the functional states of proteins, although this role remains controversial and hard to prove in living cells [4].

*PIN1* is one of the most studied and relevant prolyl isomerases in humans. This protein binds the phosphorylated serine or threonine residues preceding proline motifs (phospho-Ser/Thr-Pro) and catalyzes cis/trans isomerization of the peptide bonds [5]. *PIN1* acts on cell cycle regulator proteins, such as cyclin-dependent kinases (CDKs), and it is involved in the fine control of their functions, stability, localization, interactions and activity [5]. Therefore, *PIN1* has a central role in cell cycle progression and cancer [5]. Moreover, this isomerase has also been linked to the immune system specially in promoting inflammation and reactive-oxygen species (ROS) [5].

Spermatogenesis consists of a series of developmental steps that comprise the differentiation of a spermatogonia into a mature spermatozoon through mitosis, meiosis and spermiogenesis [6] regulated by the coordinated expression of many genes. Specifically, the generation of viable spermatozoa involves complex protein interactions and the activation of specific biological pathways that rely on a correct protein structure and activity [6]. Interestingly, *Pin1* is highly expressed in adult mice testis, particularly in spermatogonia and Sertoli cells [7,8]. It has been shown that *Pin1* knockout mice (*Pin1^−/−^*) are able to complete spermatogenesis in their early life but suffer a progressive spermatogonial stem cell (SSC) loss with age [7,8]. Indeed, it has been proposed that Pin1 is required to control the proliferation, survival and cellular commitment of undifferentiated spermatogonia by promoting mitosis in this cell lineage [7,8]. Additionally, Sertoli cells in *Pin1^−/−^* mice showed a reduced expression of N-Cadherin, a central protein in the blood–testis barrier (BTB) tight-junction system [9]. Thus, Pin1 has a role in controlling the integrity of the BTB, which is essential to maintain the immune privilege of the testis and to prevent a self-attack of the immune cells to the male germline [9].

Despite all the previously described connections with male infertility, mutations in *PIN1* have not yet been described in human male infertility cases. Male infertility affects approximately 7% of men and represents an economic, social and psychological burden for couples that seek biological parenthood all over the world [10]. Regarding male infertility, most cases are due to defects in the spermatogenic process or SPGF. The most severe forms of SPGF comprise cases with very few or no spermatozoa in the ejaculate due to non-obstructive causes: severe oligospermia (SO) or non-obstructive azoospermia (NOA), respectively [10,11]. The histological profiles of SPGF are heterogeneous and not all patients can benefit from testicular sperm extraction (TESE) and subsequent in vitro fertilization (IVF) techniques [11]. Furthermore, a molecular cause for infertility can only be established for around 25% of the NOA cases [12]. In the remaining patients, the nature of the infertility is classified as idiopathic [12]. Unfortunately, these infertile men usually undergo testicular biopsy without a reliable estimation of the TESE success. There is growing evidence that the etiology of idiopathic NOA cases might be complex and that common genetic variants in the human genome might contribute to this condition [13].

Considering the above, this study aimed to address the association between human SPGF and common genetic variants located in the *PIN1 locus* in a considerably large cohort of European infertile men.

## 2. Materials and Methods

### 2.1. Patients and Clinical Definition

This study comprises the largest SPGF cohort with European ancestry recruited for a genetic association study to date. The sample set included 715 SPGF cases from the Iberian Peninsula, who were classified as either SO (*n* = 210) or NOA (*n* = 505) as described elsewhere [14,15]. A geographically matched population with similar ethnicity and age was used as a control. This control set reached 1058 men, 358 of them were unaffected and 700 were representative of the male population (with, at least, one self-reported biological child).

SO and NOA were clinically identified according to the guidelines of the World Health Organization [16] in public hospitals and private clinics in Spain and Portugal. These conditions were diagnosed by the presence of <5 million spermatozoa/mL semen (SO) or no sperm in the ejaculate after two high-speed centrifugations (NOA). Since only idiopathic SPGF cases were considered in the analyses, individuals with abnormal karyotypes, chromosome Yq deletions, testicular disorders (orchitis, testis maldevelopment, bilateral cryptorchidism, bilateral varicocele and obstruction of vas deferens) or any sign of a possible obstructive cause were not included in our study. No significant age differences were observed between the different SPGF clinical subtypes. Although quantitative data regarding hormone levels and testis reduction were not available in the majority of the cases, we did not exclude patients based on these parameters as the clinicians did consider their values for diagnosis as established in the Canadian Urological Association (CUA) [17] and the American Urological Association (AUA)/American Society for Reproductive Medicine (ASRM) guidelines [18].

Due to the clinical relevance of the testicular biopsies, the samples obtained in such interventions were used for histological classification and resulted in 3 main subtypes (Appendix A): (1) hypospermatogenesis (HS), characterized by very low numbers of mature motile sperm cells in few testicular regions, (2) meiotic arrest (MA), showing a meiotic failure at >90% of their spermatogonia or primary spermatocytes and (3) Sertoli cell-only (SCO) syndrome if no germ cells were observed and only Sertoli cells were present in the seminal tubules. The successful or unsuccessful retrieval of sperm cells in the biopsies was stated as TESE+ or TESE-, respectively [13,15].

### 2.2. SNP Selection and Genotyping

The *PIN1* gene is located in a 14 kb long region in the human chromosome 19 (Figure 1), which is expressed in all the cellular subtypes, including somatic and germ cells, present in testis as shown in Guo et al. [19] (Appendix A) and represents a good candidate gene to test for genetic association with male infertility. The complete *PIN1 locus*, including both the coding sequence and the regulatory regions (±5 kp from the gene), forms a unique linkage disequilibrium (LD) block in the included in the European cohort of the 1000 Genomes Project Phase III (1KGPh3) (https://ldlink.nci.nih.gov/?tab=ldmatrix; accessed on: 15 June 2021) [20] (Figure 1, Appendix A). Three single-nucleotide polymorphisms (SNPs) were selected to address the genetic association of this locus with SPGF: rs2287839, rs2233678 and rs62105751. Two of these variants are located in the distant 5′ upstream regulatory region (URR) in the 5′ vicinity of *PIN1* promoter and the remaining variant is located in the third intron in this gene (Figure 1, note that *PIN1-DT* refers to *PIN1* divergent transcript). We applied a SNP tagging strategy as implemented in Haploview V.4.2 (Broad Institute; Cambridge, Massachusetts, USA) [21] covering all the common genetic variation (r^2^ ≥ 0.8) included in the European cohort of the 1000 Genome Project Phase III (1KGPh3) (https://www.ensembl.org/Homo_sapiens/; accessed on: 15 June 2021) [22]. Therefore, the variants are representative of different minor allele frequency (MAF) ranges: high (MAF > 0.3), medium (0.1 < MAF < 0.3) and low (MAF < 0.1).

We obtained genomic DNA from peripheral blood mononuclear cells (PBMCs) using standard DNA isolation kits and carried out the genotyping of each individual for these SNPs in a 7900HT Fast Real-Time PCR System (Applied Biosystems, Foster City, California, USA) using the TaqMan™ (Thermo Fisher Scientific, Pleasanton, California, USA) allelic discrimination technology with 3 pre-designed probes (assay IDs: C__16183184_40, C___2885187_10, C__89465150_10), as previously described [15]. The genotype call rate success was over 99% for all three genetic variants.

### 2.3. Statistical Analyses

The statistical power of the study cohort to detect an association was estimated with the software Genetic Association Study (GAS) Power Calculator (https://doi.org/10.1101/164343; accessed on: 15 June 2021) (Appendix A). Considering a strong allelic effect (OR ≥ 1.5), the recruited cohort reached a statistical power higher than 80% for the 3 selected polymorphisms. A possible deviation from Hardy–Weinberg equilibrium (HWE) was evaluated using a Χ^2^ test considering a 5% significance level.

Case–control comparisons of the allele and genotype frequencies were performed between the established SPGF groups and the control population. Moreover, NOA cases with and without a specific phenotype/TESE outcome were also compared to remove the disease as a possible confounding factor. The threshold for statistical significance was set at *p*-value < 0.05 after multiple testing corrections by Benjamini and Hochberg False Discovery Rate (FDR) [23]. We calculated the corresponding odds ratios (ORs) and 95% confidence intervals (CI) for all the analyses. The additive, dominant, recessive and genotypic (2 degrees of freedom) association models were tested based on logistic regression using PLINK version 1.9. (https://www.cog-genomics.org/plink/1.9/credits accessed on: 15 June 2021 [24]. To control for a possible geographical origin effect, the country of origin (Spain or Portugal) was included as a covariate in all the analyses.

As mentioned above, the three analyzed polymorphisms belong to the same haplotype block. Therefore, haplotype-based logistic regression tests were performed with geographical origin included as a covariate. Allelic combinations showing a MAF < 0.01 were not considered in these analyses. In order to confirm the contribution of each SNP to the significance of the genetic association compared to the haplotypes, a likelihood ratio test was conducted in which the haplotype model was tested against each independent SNP model.

Finally, we tested the independence between the studied polymorphisms by conditional logistic regression analyses as implemented in the PLINK software [24].

### 2.4. In Silico SNP Functional Characterization

We took advantage of the large variety of public databases and resources that provide functional evidence to prioritize variants and propose a putative molecular mechanism for the observed associations. We extended our in silico SNP functional characterization not only to the genotyped SNPs but also to all their proxies (genetics variants showing a LD *r^2^* ≥ 0.8 in the European subpopulation included in the 1000 Genomes Project) as implemented in LDlink (https://ldlink.nci.nih.gov/?tab=ldmatrix; accessed date: 13 January 2022) [20]. Genomic coordinates for all the reported variants and regions correspond to the GRCh38 human genome build.

The role of the different polymorphisms as cis expression and/or splicing quantitative trait loci (QTL), eQTL and sQTL, respectively, was obtained from the v8 GTEx data release (https://www.gtexportal.org/ accessed on: 25 July 2021) [25]. We prioritized those variants with a QTL effect in the testis. Their locations in the regulatory regions in testis were defined by overlap with testis specific assays in ENCODE [26]: DNase-seq hypersensitivity sites (ENCFF323BCL, ENCFF608KRZ); CTCF (ENCODE sample references: ENCFF300WML, ENCFF559LDF, ENCFF644JKD, ENCFF767LMP, ENCFF788RFY, ENCFF855EVV) and POLR2A (ENCFF535DHF, ENCFF651APG) protein ChIP-seqs; and H3K4me3 (ENCFF286DAB, ENCFF509DBT), H3K4me1 (ENCFF316MJM), H3K27ac (ENCFF610XSK, ENCFF819NRA), H3K9me3 (ENCFF711LHL) and H3K27me3 (ENCFF881OHS) histone modification ChIP-seqs. Additional functional clues per SNP were also obtained from dedicated integration databases such as SNPnexus (https://www.snp-nexus.org/, accessed on: 25 July 2021) [27], HaploReg (https://pubs.broadinstitute.org/mammals/haploreg/haploreg.php, accessed on 25 July 2021) [28] and SNP2TFBS (https://ccg.epfl.ch/snp2tfbs/ accessed on: 25 July 2021) [29]. These online tools organize the information included in: Ensembl, SIFT, Polyphen, CpG, Vista enhancers, miRbase, TarBase, TargetScan, miRNA Registry, snoRNA-LBME-DB, Roadmap, Ensembl regulatory build, CADD, DeepSEA, EIGEN, FATHMM, fitCons, FunSeq2 GWAVA, REMM and RegulomeBD [30] (Appendix A).

The online tools in the GTEx and LDmatrix portals were used for figure generation together with custom R scripts (version 4.2.0,The R foundation for Statistical Computing, https://www.r-project.org/ accessed on: 25 July 2021) [20,25].

## 3. Results

The three analyzed variants passed all the established quality control thresholds. Moreover, our cohort showed a very high statistical power to identify genetic associations in the range of previously reported common variant associations with SPGF [14,15] (Appendix A). No statistically significant deviation from HWE was observed either for the cases or the controls.

### 3.1. Testing for Association with Idiopathic Spermatogenic Failure Overall

Our analyses revealed no significant associations with SPGF neither under the additive nor under the recessive models after multiple testing corrections (Table 1, Appendix A).

Nevertheless, the less common SNP, rs2287839, which is located at the URR of this *locus*, showed a trend towards association with SPGF under the additive model that did not pass the FDR correction (*p*_addadj_ = 0.055, OR_add_ = 1.38 (1.06–1.81)) (Table 1). Moreover, we observed significant genetic associations for rs2287839 with SPGF under both the dominant (*p*_dom_ = 9.63 × 10^−3^, OR_dom_ = 1.46 (1.10–1.94)) and the genotypic models (*p*_geno_ = 2.39 × 10^−2^), but only the dominant model association remained significant after multiple testing adjustments (*p*_domadj_ = 2.89 × 10^−2^) (Appendix A).

While the analysis of the SO group did not present strong evidence of association with this phenotype, the comparison between the NOA patients and the control group followed the same trends of association as the SPGF group and showed significant allelic effects under the additive (*p*_add_ = 7.81 × 10^−4^, OR_add_ = 1.61 (1.22–2.13)), the dominant (*p*_dom_ = 3.01 × 10^−4^, OR_dom_ = 1.72 (1.28–2.32)) and the genotypic models (*p*_geno_ = 1.01 × 10^−3^) in the case of rs2287839 (Appendix A). Additionally, carriers for the rs2287839 and rs62105751 were more frequent in the NOA subgroup than in the SO patient set (*p*_add_ = 5.86 × 10^−3^, OR_add_ = 2.08 (1.24–3.50) and *p*_add_ = 4.56 × 10^−3^, OR_add_ = 1.48 (1.13–1.94), respectively).

When the susceptibility effects of the haplotypes were tested, we observed significant associations with NOA especially for the combinations including either two or three risk or protection alleles (Appendix A). Nevertheless, we detected that none of these combinations explained the association with NOA better than rs2287839 alone (Appendix A).

### 3.2. PIN1 Polymorphisms Have a Subtype-Specific Effect in SCO

We performed further analyses in order to address the association of such polymorphisms with specific histological patterns of NOA. The classification of the NOA patients into more homogeneous groups based on clinical and pathological criteria brought to light that the observed risk association was clearly biased towards an extreme NOA subphenotype, with a total absence of the germline in the seminiferous tubules, namely the SCO subset of patients (Table 1).

In the case of the SCO group, we observed significant associations for the three selected genetic variants under the additive model (OR_add_rs2287839 = 1.85 (1.17–2.93), OR_add_rs2233678 = 1.62 (1.11–2.36), OR_add_rs62105751 = 1.43 (1.06–1.93)), which remained significant even after multiple testing corrections (Table 1). Furthermore, the minor alleles showed strong risk effects for this subset in the comparison against the control group, i.e., all the observed ORs were greater than OR = 1.4 (OR_add_rs2287839 = 1.85 (1.17–2.93), OR_add_rs2233678 = 1.62 (1.11–2.36), OR_add_rs62105751 = 1.43 (1.06–1.93)) (Table 1). It should be noted that we also observed an increased frequency in SCO compared to the rest of the NOA subsets (Table 1).

There were significant differences between the genotype distributions for the three SNPs (*p*_geno_rs2287839 = 2.24 × 10^−2^, *p*_geno_rs2233678 = 2.93 × 10^−2^, *p*_geno_rs62105751 = 3.87 × 10^−2^), but only the effects of rs2233678 and rs62105751 remained significant after multiple testing corrections were applied (Appendix A). However, the best fitting inheritance model was the recessive ones for the most frequent variants, rs2233678 and rs62105751, and the dominant model for rs2287839 (Appendix A).

The haplotype analyses revealed that the combination of risk or protection alleles was associated with the SCO group (Appendix A). However, in this case, all the polymorphisms and the haplotype explained a similar proportion of the phenotypic variance (Appendix A). Only the rarest minor allele, rs2287839-G, showed a significant association with the group of individuals with an unsuccessful sperm retrieval during TESE (*p*_add_ = 4.19 × 10^−2^, OR_add_ = 1.55 (1.02–2.37), *p*_dom_ =2.05 × 10^−2^, OR_dom_ = 1.70 (1.08–2.65), data not shown), which did not reach the significance level after multiple testing corrections.

Considering that there is no recombination hotspot in the *PIN1 locus* (Figure 1), the possibility of one versus several independent association signals was explored. We performed multiple logistic regression analyses that combined the tested variants in pairs (Appendix A). Our results showed that all the variants lost significance when conditioned (Appendix A), thus reflecting that there is no independence between them and that they tag the same association signal.

### 3.3. In Silico Data from the GTEx Repository Suggest That the Genetic Variants in the PIN1 Locus Have Regulatory Functions on Gene and Isoform Expression

Our experimental design allowed us to study the influence of the genetic variants located in the *PIN1 locus* on the susceptibility to male infertility and specially to identify their contribution to SCO as an etiologic factor. The genetic association tests highlighted the role of lower frequency variants in the predisposition to complete lack of sperm cells in the testicles as described above. Therefore, following this lead we computationally analyzed the functional evidence and predicted effects for all the genetic variants in this locus to prioritize the most likely causal variants.

As depicted in Figure 1, the selected genomic region in chromosome 19 includes both the full *PIN1* gene and a *PIN1* divergent transcript, which has been characterized as a long non-coding RNA (lncRNA *PIN1-DT*, ENSG00000267289.1). *PIN1* is highly expressed in the testis (Figure 2). Considering that no SNP in the coding region of *PIN1* was tagged by the associated variants, we focused on the genetic variants that have been described to affect the mRNA expression (eQTLs) or the mRNA splicing (sQTLs) of this gene in the GTEx project [25]. Up to 38 SNPs overlapped with testis-specific assays in the ENCODE database (https://www.encodeproject.org/; data obtained in 22 April 2020) [26] and have been characterized as *PIN1* sQTLs (36 SNPs) or as both *PIN1* eQTLs and sQTLs (2 SNPs) in the GTEx repository testicular tissue samples. In total, 4 out of these 38 SNPs encoded DNA sequence changes that were predicted to affect spermatogenesis-related transcription factors (Appendix A).

Among the prioritized variants, the rs3810166 SNP held the strongest evidence of functionality within this locus. The minor allele of this SNP, rs3810166-G, is a proxy of the observed rs2287839-G risk allele and, according to the GTEx dataset (which includes 322 individuals), it correlates with a decreased expression of *PIN1* (Figure 2A) and it alters the *PIN1* isoform balance in the testis (Figure 2B,C). The rs3810166 SNP is almost in complete linkage disequilibrium (*r^2^*_Europeans_ = 0.94) with the top GTEx eQTL variant in the testicular tissue (rs138970490), with the magnitude of the reported effect (normalized effect size, NES = −0.25) corresponding to a log allelic fold change = −0.14 (i.e., a 10% decrease in *PIN1* expression), which is very relevant. The individuals included in the GTEx project were healthy controls, but we hypothesize that the pathogenic effects of such changes might even be stronger in the SCO context of gene expression deregulation. Although this variant is located upstream of the *PIN1* gene, it is enriched with chromatin activity, histone marks and CTCF binding sites, which supports the role of this region in the control of the expression of nearby genes (Appendix A). Additionally, the minor allele of this SNP is predicted as highly damaging by multiple functional consequence prediction algorithms and it is also predicted to alter the binding of both HDAC2, a key histone deacetylase in cell cycle progression [31], and NRSF [32], a very relevant transcriptional regulator (Appendix A). An additional prioritized SNP, rs10410379, was tagged by rs2287839 and predicted to decrease the binding of HDAC2.

The minor alleles of the two remaining polymorphisms, rs28802413 and rs10425775, have been described to affect the isoforms of *PIN1* as sQTLs. They were also linked to rs2287839 and predicted to affect the binding of relevant transcription factors in the spermatogenic process such as SIN3A [33,34] and NANOG [35] (Appendix A).

## 4. Discussion

The analysis of common variants located in potentially relevant genes for spermatogenesis is crucial for unraveling the genetic component underlying the male infertility phenotypes with idiopathic molecular etiology [13]. Frequent mutant alleles are known to modify or disrupt gene expression subtly and, in specific contexts or if certain environmental factors are present, they might alter the correct production of male gametes [36,37]. Therefore, the reported findings should not be interpreted as highly damaging and rare mutations causing infertility, but in the context of the identification of genetic markers for a complex disease [13].

In the present study, genetic association analyses on DNA from a large and clinically well-characterized cohort were performed. Additionally, we carried out a deep in silico characterization of the prioritized polymorphisms in the *PIN1 locus*. Since there are no recombination hotspots in this region (Figure 1), the LD between the selected SNPs corresponded to a high D’ and a low *r^2^* between the variants (Appendix A). We genotyped three SNPs that tag polymorphisms located in the same haplotypic block but with different MAF ranges in order to maximize the coverage of the tagging strategy.

Our results emphasized the role of PIN1 as a risk locus for male infertility. We observed associations of all the tested variants with SCO, the most severe form of NOA (Table 1). Moreover, the risk effects were also significant in the haplotype analyses (Appendix A). This phenotype is characterized by the complete absence of germ cells in the testis, in which the tubules are only composed of Sertoli cells [38]. Sertoli cells provide physical and nutritional support for the germ cells during spermatogenesis, and they form a physical barrier that prevents the immune cells from attacking the germline [39]. The present report discusses for the first time the potential association of *PIN1* with SPGF, but remarkably thanks to our comprehensively characterized patient cohort, we were also able to test for subtype specific associations. In fact, the association signals observed in the SPGF, NOA and TESE- groups of patients were likely originated by the association observed in the SCO subtype. The SCO subset of patients is histologically more homogeneous and harbors the largest risk effect sizes. Therefore, it is likely that the presence of SCO patients in the SPGF, NOA and TESE- groups is responsible for the observed associations in these groups of patients.

Moreover, the association of the three selected variants, as well as the mutual dependence between them, provided us with compelling evidence for a common causal variant or variants underlying the association signal found in this region (Appendix A). Although the most associated tag-variant, i.e., rs2287839, is located near epigenetic marks in several tissues, it is not predicted to alter the binding of relevant transcription factors in spermatogenesis. However, the strong LD patterns within the region might point towards rs2287839 as a good proxy for a putative causal risk haplotype located in the vicinity. Additionally, our results may also indicate that the causal variant/s in this locus would be in the lowest MAF ranges, as the largest effect sizes in different subsets of patients corresponded to rs2287839, which had the lowest frequency among the tested variants (MAF = 0.06).

As we aimed to prioritize plausible causal variants in this locus, we integrated a variety of functional genomic tools and datasets in an in silico approach, which led to the identification of rs3810166 as a plausible causal variant for the *PIN1* association with SCO. The rs3810166 SNP seems to affect both the expression levels and the isoform balance of *PIN1* in the healthy control samples of the GTEx repository. The minor allele of rs3810166-G decreases the expression of *PIN1* in the testis tissue and it could additionally perturb the equilibrium between the most frequent *PIN1* isoform, ENST0000247970.8, and a shorter transcript with an alternative promoter, ENST00000591777.1, which is the second most frequent *PIN1* transcript in this tissue (Figure 2B,C). The alternative allele of this SNP disturbs predicted canonical binding sites of transcription factors that are essential for the maintenance of the BTB, such as NRSF [32], and SSC maintenance, such as HDAC2 [31] (Appendix A). In this functional prioritization, we found further evidence that might support variants in *PIN1* as genetic risk factors to suffer from spermatogonial depletion and to develop male infertility with a SCO. Three polymorphisms tagged by the genotyped variants overlapped testis-specific epigenetic assays and showed *PIN1* sQTL effects. Additionally, they were also predicted to affect the binding of other relevant transcription factors. Remarkably, the alternative allele of rs28802413 putatively influences the binding affinity of SIN3A, a key transcription factor for SSC survival (Appendix A). In line with this, the lack of *Sin3a* expression in mice Sertoli cells caused a microenvironment change and the loss of undifferentiated spermatogonia [33]. Furthermore, specific genetic inactivation of the *Sin3A* gene in the germline of adult testes led to a SCO phenotype in mice. The loss of *Sin3A* expression causes apoptosis of the germ cells, since they require the correct function of the Sin3–HDAC complex, but it also alters the gene expression profile in Sertoli cells [34] (Appendix A). The analysis of the Sertoli cell transcriptome has revealed that NANOG, another transcription factor that might be affected by genetic variation in *PIN1* (Appendix A), is expressed in Sertoli and interstitial cells of neonatal testes (but not in the adult testes) and also in Sertoli cells from SCO patients [35].

The effect of some of these polymorphisms or a combination of them would eventually imbalance the expression of *PIN1*, which has been associated with male infertility in animal models. *PIN1* is expressed in all the cell stages of the differentiating male germline and in Sertoli cells (Appendix A) it has a prominent role in chromosome segregation [40] and it interacts with the androgen receptor (AR) [41]. Knockout mouse models have shown that the genetic silencing of Pin1 causes spermatogonial depletion [7], affects the Sertoli cells and disturbs the BTB [9]. Additionally, it causes a progressive loss in the progression of the mitotic cell cycle of SSC in steady state [8] and deregulates the timing of primordial germ cell proliferation during the embryonic development [42]. *PIN1* has also been pinpointed as a seminal biomarker for higher fertility in porcine models [43]. Finally, we wish to highlight a recent study where the intracellular injection of *PIN1* lipid nanoparticles in knockout mice (*Pin1^−/−^*) resulted in the regulation of spermatogonial proliferation and differentiation and in the restructuring of the BTB, thus rescuing fertility in male mice [44]. Therefore, we consider that our findings encourage the analysis of *PIN1* as a therapeutic target to restore human male fertility.

## 5. Conclusions

Altogether, the selection of *PIN1* as a candidate genetic risk factor for SPGF in humans and the analysis of common variation proved to be a successful strategy. We hypothesize that the nature of the majority of the cases of SPGF classified as idiopathic is complex and polygenic, with a low individual contribution of a high number of genetic risk factors, which combined with environmental factors may lead to male infertility [45,46]. However, the interrogation of the role of common genetic variation in this heterogeneous phenotype and the analysis of more homogeneous histological subsets of patients will contribute to the knowledge of the field about the disease pathogenesis. It should be noted that these approaches might eventually advance the development of a panel of genetic markers to predict TESE success and avoid surgical interventions if the odds of finding healthy spermatozoa are low, as in the case of SCO patients [13].

## Figures and Tables

**Figure 1 jpm-12-00932-f001:**
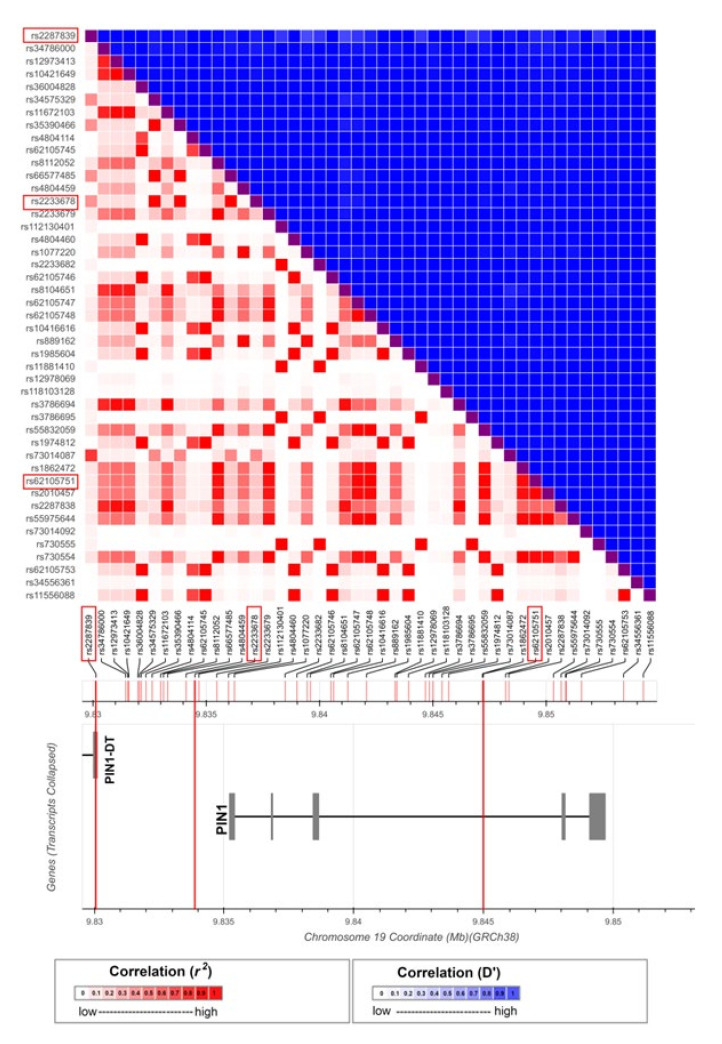
Genetic and functional structure of the *PIN1* region. Linkage disequilibrium patterns in the European population included in the 1000 Genome Project were retrieved from the LDlink repository to design a tag-SNP study for the *PIN1* locus. The selected tag-SNPs are highlighted in red. All SNP positions are reported in GRCh38. *PIN1-DT: PIN1* divergent transcript.

**Figure 2 jpm-12-00932-f002:**
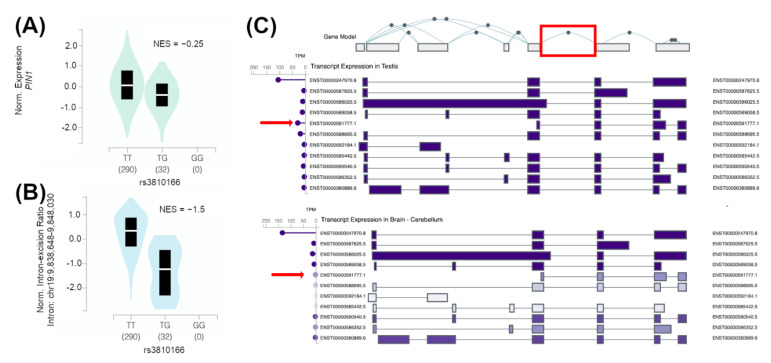
Analysis of data from the GTEx repository to detect QTL effects and isoform expression patterns in the *PIN1 locus*. (**A**) Expression-QTL (eQTL) and (**B**) splicing-QTL (sQTL) allele effects on *PIN1* of the rs3810166 variant. (**C***) PIN1* transcript expression in human testis and brain tissues. The sQTL-affected intron is highlighted in red. The transcript ENST00000591777.1 is marked with a red arrow. NES: normalized effect size.

**Table 1 jpm-12-00932-t001:** Genotype and allele frequency analyses of the tested genetic variants. The subgroups of clinical phenotypes of male infertility were compared against fertile controls under the additive model.

SNP (GRCh38 bp Position)	Alleles (1/2)	Cohort	Genotypes (11/12/22)	MAF	*p*	Adjusted *p* *	OR (CI 95%)
rs2287839	G/C	Controls (*n* = 1049)	6/129/914	0.0672	NA	NA	NA
chr19:9,830,138		SpF (*n* = 705)	4/110/591	0.0837	1.84 × 10^−2^	0.05522	1.38 (1.06–1.81)
		SO (*n* = 205)	1/17/187	0.0463	0.1741	NA	0.70 (0.41–1.17)
		NOA (*n* = 500)	3/93/404	0.099	7.81× 10^−4^	2.34 × 10^−3^	1.61 (1.22–2.13)
		SCO (*n* = 102)	1/22/79	0.1176	8.38 × 10^−3^	1.94 × 10^−2^	1.85 (1.17–2.93)
		MA (*n* = 52)	0/8/44	0.0769	0.6242	NA	1.20 (0.57–2.52)
		HS (*n* = 48)	0/10/38	0.1042	0.1453	NA	1.66 (0.84–3.28)
rs2233678	C/G	Controls (*n* = 1050)	17/206/827	0.1143	NA	NA	NA
chr19:9,834,503		SpF (*n* = 706)	13/136/557	0.1147	0.2862	NA	1.13 (0.90–1.40)
		SO (*n* = 206)	2/28/176	0.0777	0.1999	NA	0.76 (0.51–1.15)
		NOA (*n* = 500)	11/108/381	0.13	0.0784	NA	1.23 (0.98–1.55)
		SCO (*n* = 102)	5/25/72	0.1716	1.34 × 10^−2^	1.94 × 10^−2^	1.62 (1.11–2.36)
		MA (*n* = 52)	1/10/41	0.1154	0.8202	NA	1.07 (0.58–1.97)
		HS (*n* = 48)	0/11/37	0.1146	0.7795	NA	1.09 (0.58–2.07)
rs62105751	A/G	Controls (*n* = 1052)	97/468/487	0.3146	NA	NA	NA
chr19:9,847,213		SpF (*n* = 706)	72/307/327	0.3194	0.5456	NA	1.05 (0.90–1.22)
		SO (*n* = 205)	14/81/110	0.2659	0.1441	NA	0.82 (0.63–1.07)
		NOA (*n* = 501)	58/226/217	0.3413	0.102	NA	1.15 (0.97–1.36)
		SCO (*n* = 102)	17/46/39	0.3922	1.94 × 10^−^^2^	1.94 × 10^−^^2^	1.43 (1.06–1.93)
		MA (*n* = 52)	6/23/23	0.3365	0.5002	NA	1.16 (0.76–1.77)
		HS (*n* = 48)	1/23/24	0.2604	0.3656	NA	0.80 [0.50–1.29]

* *p* adjusted is from FDR_BH.

## Data Availability

The generated genotyping information is available upon formal and reasonable request to the corresponding author.

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
