# Peer review of "Common Variation in the PIN1 Locus Increases the Genetic Risk to Suffer from Sertoli Cell-Only Syndrome"

_jpm, 2022, doi:10.3390/jpm12060932_

Round 1
Reviewer 1 Report
The article “Common variation in the PIN1 locus increases the genetic risk to suffer from Sertoli Cell Only syndrome” reports common PIN1 variants could potentially impact its expression and isoform balance resulting in SCO. It is promising to show the novel patients’ data of PIN1 and the bioinformation work is solid and convincible, but I still have some questions about the roles the variants play.
Major Comments:
- What degree would the PIN1 expression decrease, or isoform balance change due to the common variations? It is easy to know by luciferase assay and qPCR.
- Homozygous knockout mice are able to complete spermatogenesis in their early age which is not consistent with the degree of SCO in human, especially for not total depletion of PIN1 caused by variants.
Minor Comments:
What’s the average age of the SCO patients with PIN1 variants? Is it significantly higher than the others?
Author Response
Reviewer 1
The article “Common variation in the PIN1 locus increases the genetic risk to suffer from Sertoli Cell Only syndrome” reports common PIN1 variants could potentially impact its expression and isoform balance resulting in SCO. It is promising to show the novel patients’ data of PIN1 and the bioinformation work is solid and convincible, but I still have some questions about the roles the variants play.
We thank the reviewer for their kind comments and positive evaluation of our work. Moreover, we appreciate the reviewer’s suggestions, which have definitely improved this version of the manuscript.
Major Comments:
-
What degree would the PIN1 expression decrease, or isoform balance change due to the common variations? It is easy to know by luciferase assay and qPCR.
We apologize with the reviewer for the lack of this information both in Figure 2 and in the manuscript text. The GTEx project provides a measure of the effect on gene and intron expression displayed as normalized effect size (NES) for all the reported QTLs. NES is the slope of the QTL linear regression, and it is computed as the effect of the alternative allele (ALT) relative to the reference allele (REF) in the human genome reference.
Since NES are computed in a normalized space, they can’t be biologically interpreted in a direct way. Therefore, additional information is required in order to estimate the real biological effect of a genetic variant on gene expression, i.e. the log allelic fold-change (aFC). The aFC is a measure of cis-eQTL effect size, and it is defined as the log-ratio between the expression of the haplotype carrying the alternative allele to the one carrying the reference allele of the polimorphism. The aFC is represented in the log2 scale and, currently, only the aFC of the top eQTL variant of each gene is available in GTEx. Nevertheless, our prioritized variant (rs3810166) is almost in complete linkage disequilibrium with the top eQTL variant for PIN1 (rs138970490) in GTEx and their NES and aFC are equivalent and relevant.
Consequently, we have updated Figure 2 to show NES information and we have modified the text to discuss it the biological implications of this parameter:
Page 10: “Among the prioritized variants, the rs3810166 SNP held the strongest evidence of functionality within this locus. The minor allele of this SNP, rs3810166-G, is a proxy of the observed rs2287839-G risk allele and, according to the GTEx dataset (which includes 322 individuals), it correlates with a decreased expression of PIN1 (Figure 2A) and it alters the PIN1 isoform balance in the testis (Figure 2B-C). The rs3810166 SNP is almost in complete linkage disequilibrium (R2Europeans = 0.94) with the top GTEx eQTL variant in the testicular tissue (rs138970490), with the magnitude of the reported effect (normalized effect size, NES = -0.25) corresponding to a log allelic fold-change = -0.14 (i.e. a 10% decrease in PIN1 expression), which is very relevant.”
In order to answer a concern raised by Reviewer 2, we also added the following sentence in page 10:
“The individuals included in the GTEx project were healthy controls, but we hypothesize that the pathogenic effects of such changes might even be stronger in the SCO context of gene expression deregulation.”
-
Homozygous knockout mice are able to complete spermatogenesis in their early age which is not consistent with the degree of SCO in human, especially for not total depletion of PIN1 caused by variants.
We agree with the reviewer that the biological effects of the reported variants can’t be held accountable for SCO on their own. In fact, as it is stated in the manuscript (page 3) and the reviewer pointed out, although the spermatogonial cells died in the long-term, knock-out Pin1-/- mice models were able to complete the spermatogenic process in their early life. Therefore, a correct control of PIN1 expression is likely to be required to keep testicular homeostasis and to maintain fertility, but the mechanisms underlying the start of the testicular disruption in this animal model are yet to be elucidated. In fact, in the recruited human cohort included in our study, we analyzed the association of PIN1 genetic polymorphisms with SCO susceptibility and we showed that the minor alleles of these polymorphisms were more frequent in SCO patients than in matched controls. But some fertile men did carry PIN1 risk alleles. Moreover, we also found evidence showing that the effects of these genetic variants on PIN1 expression are relevant, but the risk alleles lead to a reduced expression of PIN1 and not to a total silencing of the gene.
Consequently, we think that it is essential that the observed results in the PIN1 locus are considered in the light of recent findings that suggest that unexplained SPGF is a complex disease (Cerván-Martín et al. 2020, Journal of Clinical Medicine). The genetic basis of complex diseases is not characterized by a small number of very penetrant rare mutations. On the contrary, genetic risk loci for complex diseases comprise tens or hundreds of polymorphisms located in different loci, which modestly increase the genetic predisposition to suffer from the disease (Ghosh & Collins, 1996. Ann Rev Med; Visscher et al. 2017. Am J Hum Genet). Then, the effect of often unknown environmental factors on genetically predisposed individuals would finally trigger the disease (Visscher & Goddard, 2019, Genetics). Therefore, we worked under the biological hypothesis that PIN1 risk variants modestly contribute to increasing the risk of men to develop SCO, but they can’t be accounted for as the only genetic markers for these genetically predisposed affected men.
Following the reviewer’s advice, and in order to clarify the relevance of our findings, we have included the following statement in the discussion section of the revised version of the manuscript:
“Therefore, the reported findings should be interpreted not as highly damaging and rare mutations causing infertility, but in the context of the identification of genetic markers for a complex disease [13].”
Minor Comments:
What’s the average age of the SCO patients with PIN1 variants? Is it significantly higher than the others?
We observed no significant age at diagnose differences between SCO and non-SCO patients (AgeSCO = 33.42 years, AgeNon-SCO = 35.22 years). We have added the following sentence in the Methods section:
Page 6: “No significant age differences were observed between the different SPGF clinical subtypes.”
Reviewer 2 Report
The Cerván-Martín et al., 2022, Manuscript ID: jpm-1713323 addresses the role of PLIN1 gene common genetic variants required to control the proliferation of spermatogonial stem cells and the integrity of the blood-testis barrier, in the genetic risk of 68 developing male infertility due to a severe spermatogenic failure. A search on Pubmed.gov for the terms "PIN1" and "Sertoli" keywords resulted in only 3 hits that depicts the novelty of this study.
The authors have planned nicely this study but still there are few queries and few suggestion which makes this manuscript more representable to be publish.
- In the supplementary figure 1, no Leydig cells population is there, but in the original paper Guo et al., that population is listed as Leydig cells. Please justify if not proper marker has been used in study?
- Do the authors have any plan to conduct the future research to know the molecular mechanisms of beneficial role of PIN1 in the Sertoli cell only syndrome?
- Do the authors see any changes in the mRNA and protein level changes in the expression of PIN1 during Sertoli cell only syndrome or infertile pateints?
Author Response
Reviewer 2
The Cerván-Martín et al., 2022, Manuscript ID: jpm-1713323 addresses the role of PLIN1 gene common genetic variants required to control the proliferation of spermatogonial stem cells and the integrity of the blood-testis barrier, in the genetic risk of 68 developing male infertility due to a severe spermatogenic failure. A search on Pubmed.gov for the terms "PIN1" and "Sertoli" keywords resulted in only 3 hits that depicts the novelty of this study.
We are grateful to the reviewer for their consideration of the novelty of the results shown in this manuscript. We share the reviewer’s point of view that this report will be of interest for the community and that it may eventually lead to an improvement in patient classification and in patient prognosis and clinical management.
The authors have planned nicely this study but still there are few queries and few suggestion which makes this manuscript more representable to be publish.
-
In the supplementary figure 1, no Leydig cells population is there, but in the original paper Guo et al., that population is listed as Leydig cells. Please justify if not proper marker has been used in study?
The reviewer noticed a lack of correspondence between the cluster labels and the legend in the adapted figure from Guo et al. We sincerely apologize for this mistake, which has been corrected in the revised manuscript (Supplementary Figure 1 has been updated).
-
Do the authors have any plan to conduct the future research to know the molecular mechanisms of beneficial role of PIN1 in the Sertoli cell only syndrome?
Considering the expression of PIN1 in the different cell types in human testis (Supplementary Figure 1) and the promising outcomes of Pin1 injection in KO mouse models (page 12), we share the reviewer’s expectations about the clinical benefits of PIN1. Additionally, there is firm evidence regarding PIN1 interaction with key cell cycle regulators, such as TP53 or MYC (page 3 and Figure A below). Unfortunately, there is no ongoing clinical trial in humans, neither involving small molecule binding nor PIN1 injection.
Nevertheless, our future research plans include a whole transcriptome analysis at the single cell level which will include a number of patients classified into different SPGF clinical subphenotypes, including SCO and, of course, address PIN1 expression. However, this project is still in the early phases and we don’t have any data to support a more detailed functional characterization of PIN1 role in SCO patients yet.
Figure A. Experimental and co-expression evidence of protein-protein interactions between PIN1 and others. Data obtained from STRING [Szklarczyk et al. 2015. Nucleic Acids Res.]
-
Do the authors see any changes in the mRNA and protein level changes in the expression of PIN1 during Sertoli cell only syndrome or infertile pateints?
Unfortunately, there is no current large expression dataset available for SCO patients which includes PIN1 and which has been also genotyped for the selected polymorphisms. However, as we mentioned in our response to Reviewer 1, we are positive about the impact of the reported variants in PIN1 gene expression due to the GTEx project dataset analyses in healthy controls:
“The GTEx project provides a measure of the effect on gene and intron expression displayed as normalized effect size (NES) for all the reported QTLs. NES is the slope of the QTL linear regression, and it is computed as the effect of the alternative allele (ALT) relative to the reference allele (REF) in the human genome reference”.
Since NES are computed in a normalized space, they can’t be biologically interpreted in a direct way. Therefore, additional information is required in order to estimate the real biological effect of a genetic variant on gene expression, i.e. the log allelic fold-change (aFC). The aFC is a measure of cis-eQTL effect size, and it is defined as the log-ratio between the expression of the haplotype carrying the alternative allele to the one carrying the reference allele of the polimorphism. The aFC is represented in the log2 scale and, currently, only the aFC of the top eQTL variant of each gene is available in GTEx. Nevertheless, our prioritized variant (rs3810166) is almost in complete linkage disequilibrium with the top eQTL variant for PIN1 (rs138970490) in GTEx and their NES and aFC are equivalent and relevant.
Consequently, we have updated Figure 2 to show NES information and we have modified the text to discuss it the biological implications of this parameter:
Page 10: “Among the prioritized variants, the rs3810166 SNP held the strongest evidence of functionality within this locus. The minor allele of this SNP, rs3810166-G, is a proxy of the observed rs2287839-G risk allele and, according to the GTEx dataset (which includes 322 individuals), it correlates with a decreased expression of PIN1 (Figure 2A) and it alters the PIN1 isoform balance in the testis (Figure 2B-C). The rs3810166 SNP is almost in complete linkage disequilibrium (R2Europeans = 0.94) with the top GTEx eQTL variant in the testicular tissue (rs138970490), with the magnitude of the reported effect (normalized effect size, NES = -0.25) corresponding to a log allelic fold-change = -0.14 (i.e. a 10% decrease in PIN1 expression), which is very relevant.”
Therefore, we consider that these effects will be very likely maintained or even increased in SCO patients due to the pathogenic environment in affected testes. This comment has been included in page 10:
“The individuals included in the GTEx project were healthy controls, but we hypothesize that the pathogenic effects of such changes might even be stronger in the SCO context of gene expression deregulation.”

Round 2
Reviewer 1 Report
All the points have been well addressed.
Author Response
We thank the reviewer for their comments and suggestions through all the process as they contributed to greatly improve the quality of the manuscript.